# Direct CVD Growth of Transferable 3D Graphene for Sensitive and Flexible SERS Sensor

**DOI:** 10.3390/nano13061029

**Published:** 2023-03-13

**Authors:** Su Han Kim, Shiru Jiang, Sang-Shin Lee

**Affiliations:** 1Nano Device Application Center, Kwangwoon University, Seoul 01897, Republic of Korea; 2Department of Electronic Engineering, Kwangwoon University, Seoul 01897, Republic of Korea

**Keywords:** graphene, CVD, direct growth, SERS, sensor

## Abstract

Three-dimensional (3D) graphene (Gr) has been successfully grown on a patterned sapphire substrate (PSS) with very low mismatch between Gr and the sapphire nanostructure through metal-catalyst-assisted chemical vapor deposition (CVD). However, the transfer of the 3D Gr film without compromising the structural integrity of Gr is challenging because of the low etching rate of PSS. For easy and high-quality transfer of 3D Gr, we propose to coat a transfer-support layer (TSL) on PSS before direct CVD growth of 3D Gr. The TSL is directly deposited on PSS by atomic layer deposition without causing any structural changes in the substrate, as verified through atomic force microscopy (AFM). Few-layer 3D Gr is conformally produced along the surface of the TSL/PSS and successfully transferred onto a flexible substrate through wet-etching transfer, as confirmed by scanning electron microscopy, AFM, and Raman spectroscopy studies. We also present the fabrication of a sensitive and flexible surface-enhanced Raman scattering sensor based on 3D Gr on PMMA with high detection performance for low concentrations of R6G (10^−9^ M). The proposed transfer method with TSL is expected to broaden the use of 3D graphene in next-generation device applications.

## 1. Introduction

Three-dimensional (3D) nanostructured graphene (Gr) exhibits excellent material properties and structural advantages, making it a promising material for next-generation device applications such as electrochemical sensors, biosensors, optical sensors, and energy storage systems [1,2,3,4,5,6,7,8,9,10,11]. In a previous work, we studied the growth of 3D Gr on a patterned sapphire substrate (PSS) via Cu-vapor-assisted chemical vapor deposition (CVD) [1]. A Cu foil was sandwiched between two PSSs and placed at the center of a quartz tube. The Cu vapor generated during high-temperature annealing of the foil assisted the growth of Gr having a low mismatch with the 3D structure of the PSS. We synthesized multilayer Gr with an embossed structure and identified the Dirac point of the direct-grown 3D Gr. In addition, a 3D Gr-based field-effect transistor-type pressure sensor was fabricated, which exhibited significantly enhanced sensing performance owing to the embossed structure of 3D Gr on PSS.

Recently, significant research progress has been made in the field of flexible and wearable electronic devices by transferring 3D Gr (with exceptional properties) onto flexible substrates, enriching the applications of 3D Gr [2,3,10]. Yang et al. developed a flexible and highly sensitive capacitive pressure sensor based on a micro-structured Gr electrode [12]. The micro-structured Gr electrode improved the reactivity (3.19 kPa^−1^) of the capacitive pressure sensor, and its reactivity could be tuned by tailoring the microstructure. Jing et al. fabricated flexible 3D Gr-nanowire-based thermal interface materials with an ultralow thermal resistance and excellent solderability and mechanical compliance [13].

There was a previous attempt to transfer directly grown 3D Gr onto a flexible substrate using the polydimethylsiloxane stamping method; however, the transfer yield was poor [1]. A wet-transfer method using a polymethyl methacrylate (PMMA) passivation layer and sapphire etching solution has been suggested for high-quality transfer of 3D Gr grown on PSS [14]. However, this transfer process can damage the Gr layer, especially nanopatterned 3D Gr, owing to the low etching rate of sapphire [15]. Therefore, a new approach is required to achieve high-quality transfer of directly grown 3D Gr.

In this work, we deposited a transfer-support layer (TSL) on the PSS for easy transfer of the grown 3D Gr. Multilayer 3D Gr was grown on the TSL/PSS through Cu-vapor-assisted CVD and successfully transferred onto a flexible substrate through wet etching, without causing any structural damage. We then fabricated flexible surface-enhanced Raman scattering (SERS) sensors using this flexible 3D Gr on PMMA, which could detect very low concentrations of R6G (down to 10^−9^ M). The present study is expected to provide a new strategy for the development of 3D Gr-based biosensors.

## 2. Materials and Methods

### 2.1. Direct Growth of Gr on TSL-Deposited PSS

To avoid the damage of Gr during the transfer process [15,16], a TSL that satisfies the following conditions is required: (1) excellent thermal stability at temperatures up to 1000 °C, (2) chemical stability with the gases used in Gr synthesis (CH_4_, H_2_, and Cu), and (3) fast etching rate without damaging Gr. HfO_2_ was chosen as the TSL material as it meets the above conditions [17,18,19]. A 40-nm-thick HfO_2_ film was deposited on PSS as a TSL by atomic layer deposition (ALD). Gr was then directly grown via Cu-vapor-assisted CVD. A 25-μm-thick Cu foil (99.999%, Alfa Aesar, Haverhill, MA, USA) was placed on TSL/PSSs with a small gap and the sample was then positioned at the center of a quartz tube. The sample was annealed at 1000 °C under an H_2_ flow of 50 sccm for 30 min. Subsequently, CH_4_ gas (15 sccm) was introduced to grow Gr, and the chamber was maintained at atmospheric pressure and 1000 °C for 5 min. Finally, the CVD chamber was cooled rapidly to room temperature.

### 2.2. Fabrication and Characterization of Flexible 3D Gr SERS Sensor

To support the embossed structure, the grown 3D Gr was coated with a thick (~1.3 μm) passivation layer made of 495 PMMA C9 (Microchem, Newton, MA, USA) and then transferred using the wet-transfer method. Before transferring the Gr, the side of the sample was polished with a knife to facilitate penetration of the etching solution. Then, the sample was immersed in a 1:10 HF solution to etch the HfO_2_ layer and separate the PMMA/Gr layer from the PSS. The PMMA/Gr layer was rinsed with deionized water to remove any etchant impurities. Finally, the flexible 3D Gr substrate was obtained by drying at room temperature. The semicontinuous Ag layer with a thickness of 8 nm was decorated on the 3D Gr surface via thermal evaporation in accordance with previously reported conditions [20], thus enhancing the sensing performance of SERS via Ag nanoparticles-induced electromagnetic field and the resonance energy transfer between Ag and Gr films. The morphological properties of 3D Gr were characterized using scanning electron microscopy (SEM; JEOL JSM-7600, JEOL Ltd. Tokyo, Japan) and atomic force microscopy (AFM; XE100, PSIA, Suwon, Korea).

The R6G solutions with different concentrations ranging from 10^−5^ to 10^−9^ were coated on the corresponding substrates with the intention of characterizing the SERS performance through a Raman microscopy system. In the case of the rigid 3D Gr SERS sensor, the measurement sequence of different concentrations of R6G was not necessary because the SERS sensors were independent of each other. However, only one specific sample of flexible 3D Gr SERS was used repeatedly for all concentrations of R6G solutions to obtain the Raman spectra. In order to avoid the influence of the previous measurement on the following measurement as a result of the reagent residue, the flexible SERS was cleaned in deionized water after each measurement. For the same purpose, the concentration of R6G solution was progressively increased to perform the measurement.

## 3. Results and Discussion

### 3.1. Direct Growth of 3D Gr on TSL-Deposited PSS

Figure 1a shows the schematic diagram of the direct growth of 3D Gr on HfO_2_-deposited PSS. The HfO_2_ film (40 nm thick) is uniformly deposited on the PSS by ALD [21]. Gr is then directly grown on HfO_2_/PSS via Cu-vapor-assisted CVD, as shown in Figure 1b.

### 3.2. Surface Analysis of 3D Gr on HfO_2_/PSS

The surface properties of Gr on HfO_2_/PSS were studied using SEM, Raman spectroscopy, and AFM; the results are presented in Figure 2. The conformal growth of few-layer 3D Gr on HfO_2_/PSS is evident from the optical microscopy and SEM images (Figure 2a–c). The AFM image of 3D Gr on HfO_2_/PSS in Figure 2d confirm that the 3D embossed structure is maintained well after HfO_2_ deposition and Gr synthesis. The Raman spectrum of the synthesized 3D conformal Gr is shown in Figure 2e. Three main peaks are observed at 1350, 1580, and 2650 cm^−1^, which correspond to the D, G, and 2D peaks, respectively, of Gr. Few-layer Gr grew well over the entire area of PSS, as confirmed by the ratio of the G and 2D peak intensities [22].

### 3.3. Transfer Process and Surface Analysis of 3D Gr

The widely implemented wet-etching method was used to transfer 3D Gr without causing structural damage [14]. The representative processes are shown in Figure 3a. After removing the TSL via etching, the free-standing 3D PMMA/Gr could be directly used as a flexible device, as shown in Figure 3b (inset). In addition, the PMMA/Gr can be easily transferred to other desired substrates for wider applications. As shown in Figure 3c, the embossed shape in the transferred 3D Gr can be clearly distinguished via AFM, indicating that the 3D structure originating from the PSS was well-copied and maintained. Considering the reduced height of the embossed shape in 3D Gr after the transfer, a certain degree of structural collapse may arise during the transfer process. The Raman spectrum in Figure 3d showed that the increase in the intensity of the D peak at 1350 cm^−1^ pointed to an increase in Gr defects during the transfer process. However, the presence of well-defined G and 2D peaks in Raman spectrum confirms the successful transfer of the few-layer Gr [23].

### 3.4. SERS Performances of Flexible and Rigid Sensors

The SERS scheme enables ultrasensitive detection up to the single-molecule level [24,25,26], and 3D Gr has been demonstrated to be a promising high-performance SERS sensor [5,27,28,29]. This is attributed to the ideal Raman spectrum obtained from the fluorescence-quenching characteristics of Gr [30,31] and the large areal and volumetric density of 3D nanostructure [32,33]. However, the difficulty in cleaning the sample without damaging the Gr limits the reuse of rigid substrate sensors.

To analyze this limitation, we conducted a comparative experiment using rigid and flexible SERS sensors based on 3D Gr. The rigid SERS sensors were fabricated with 3D Gr/HfO_2_/PSS, while flexible SERS sensors were fabricated with transferred 3D Gr/PMMA. Ag nanoparticles were decorated onto both SERS sensors to enhance their sensing performance. Several R6G reagents were prepared at concentrations ranging from 10^−5^ to 10^−9^ M. Then the prepared reagents were dip-coated on the surfaces of Gr/HfO_2_/PSS and SiO_2_/Si to examine the sensing performance, which is dominated by the corresponding synthesized 3D Gr-based rigid and flexible SERS sensors, respectively. Figure 4a,b show the schematic of the R6G detection experiment in relation to the rigid and flexible SERS sensors, respectively. In the case of rigid sensor (i.e., Gr/HfO_2_/PSS), the 3D Gr was directly coated with R6G with no transfer processes, and the SERS signal was obtained by shining light from a 532 nm laser for executing Raman analysis. Each concentration of R6G corresponds to a sensing device. To verify the feasibility of reuse of the developed flexible SERS sensor (i.e., Gr/PMMA), the R6G solutions were coated on multiple SiO_2_/Si substrates instead of the Gr surface. A flexible SERS sensor was then placed on the R6G-coated SiO_2_/Si to successively obtain the SERS signal.

The peak at 629 cm^−1^ observed in Figure 4c,d is the reference peak for R6G [34]. This clearly shows that the two SERS devices exhibit high R6G detection performance even at very low concentrations of 10^−9^ M. Especially, the flexible SERS sensor gave rise to a sufficiently high intensity even at an extremely low concentrations compared to the rigid SERS sensor, preserving a stable performance even after multiple uses. In addition, as shown in Figure 4c,d, the D peak originating from 3D Gr is only distinguished in the case of low R6G concentration because the intensity of the SERS peaks is so strong that the D peak of 3D Gr is concealed in the case of high concentration. It can be hence concluded that the development of a transfer scheme for 3D Gr can lead to the pioneering of new materials and applications enabling superior functionality, as demonstrated by the excellent performance and reusability of the flexible SERS sensor compared to the rigid one. This will open up new possibilities for the design and fabrication of flexible and high-performance electronic devices for various applications.

## 4. Conclusions

We developed a facile direct CVD method for the synthesis of transferable 3D Gr using TSL-coated PSS. The proposed Gr growth method overcomes the disadvantage of structural damage observed in traditional wet-transfer method. The TSL was chemically stable and could be removed quickly using an etching solution, facilitating transfer of large-area 3D Gr. The advantage of this approach is the direct use of 3D Gr as a flexible device. To illustrate the benefits of our study, we fabricated highly sensitive and reusable SERS sensors using 3D Gr on flexible substrates. A single flexible SERS sensor based on 3D Gr could be used for several measurements with performance comparable to that of a rigid sensor. We expect that the transfer method for directly grown 3D Gr will contribute to the performance enhancement and functional development of electrical and optical device applications.

## Figures and Tables

**Figure 1 nanomaterials-13-01029-f001:**
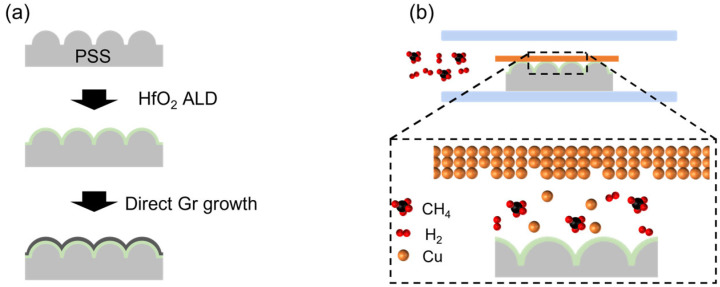
Schematic of direct growth of transferable 3D Gr on TSL-deposited PSS. (**a**) Fabrication process for transferable 3D Gr and (**b**) growth mechanism of Cu-vapor-assisted Gr.

**Figure 2 nanomaterials-13-01029-f002:**
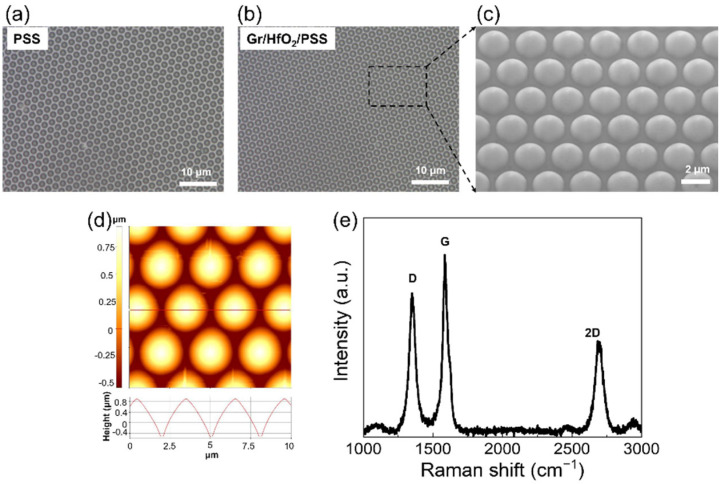
Surface analysis of 3D Gr on HfO_2_/PSS. Optical microscopy images of (**a**) PSS and (**b**) Gr/HfO_2_/PSS. (**c**) SEM image, (**d**) AFM image, and (**e**) Raman spectrum of 3D Gr on HfO_2_/PSS.

**Figure 3 nanomaterials-13-01029-f003:**
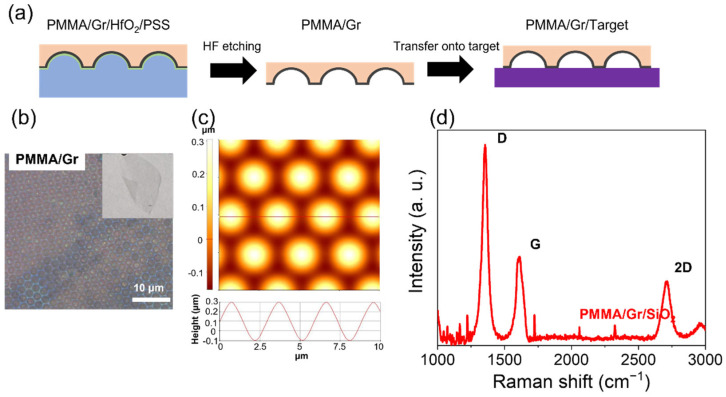
(**a**) Schematic of the transfer process of the flexible 3D Gr via wet-etching method. (**b**) Optical microscopy image of PMMA/Gr after HF etching. Inset: photograph of the flexible free-standing PMMA/Gr. (**c**) AFM image and (**d**) Raman spectrum of the transferred flexible 3D Gr.

**Figure 4 nanomaterials-13-01029-f004:**
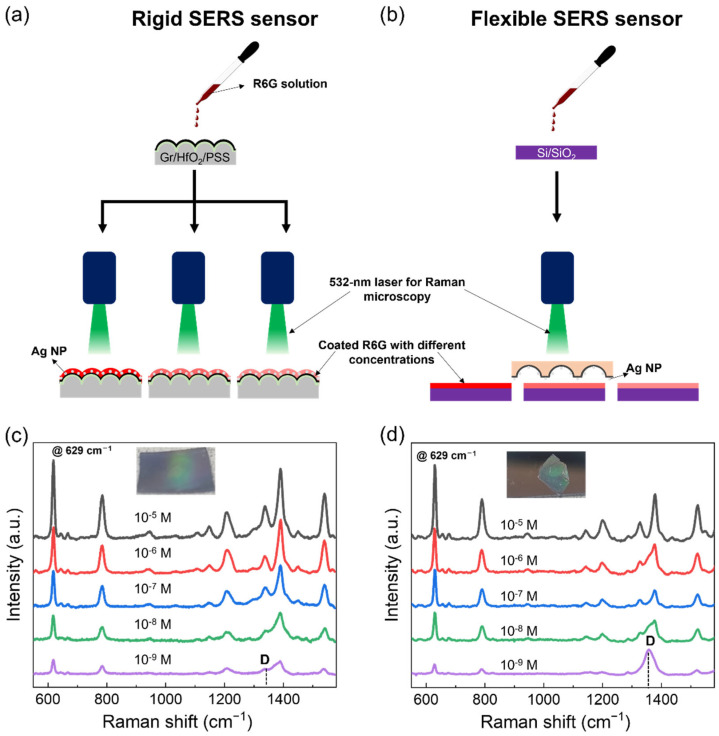
Schematic of R6G detection via SERS analysis with (**a**) rigid and (**b**) flexible SERS sensors based of 3D Gr. Raman spectra of R6G (10^−5^–10^−9^ M) on the (**c**) rigid and (**d**) flexible SERS sensors.

## Data Availability

Not applicable.

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
