# Peer review of "Direct CVD Growth of Transferable 3D Graphene for Sensitive and Flexible SERS Sensor"

_nanomaterials, 2023, doi:10.3390/nano13061029_

Round 1

Reviewer 2 Report

The manuscript “Direct CVD growth of transferable 3D graphene for sensitive and flexible SERS sensor” presents the results of the characterization and SERS measurements of the CVD growth of graphene with the supportive HfO2 layer. The manuscript is interesting and brings information about graphene growth on metal oxides. Such systems can be then transferred on other substrates or applied in e.g. electronics. Thus, the presented results are valuable and bring information on the properties of the graphene grown with the additional supportive layer.

However, several questions require broader discussion:

  1. Quality of the pictures, i.e. the AFM results (Fig. 2 d, Fig.3 c) needs to be improved, the scale is not readable.
  2. Only a single Raman spectrum is presented for each case. It is not clear, where in the sample area it was measured (on the “flat” or “convex” part). Graphene properties differ significantly with the curvature (please compare with [1]); therefore spectra collected in both areas should be presented (in the best case, Raman mapping overlapping both areas).
  3. Metal oxides are known to dope graphene [2]. A comment about the influence of the strain and doping change in graphene grown on the HfO2 layer compared with the typically grown CVD layer (e.g. using the correlation plot [3]) is needed.
  4. It is not clear what the Authors mean by the sentence “D mode indicates an increase in the asymmetry of the Gr structure” (bottom of page 3), please comment on it and add reference.
  5. The D mode, especially after the HF etching is very high. How does the graphene look like? More detailed surface analysis (AFM with higher resolution, in range 1 um, or SEM) is required. Moreover, analysis of the distance among defects is needed.
  6. SERS measurements on both types of samples are the most interesting result of the manuscript. It definitely requires more detailed discussion. For example, which factor (strain or doping) plays more important role in enhancement? The ratios of the R6G characteristic Raman peaks at 610 and 1650 cm-1 in function of the doping for both types of samples need to be presented and discussed (as eg. in Figure 8 in [4]).

[1] Drogowska-Horna, Nano Res. 13, 2332–2339 (2020) 10.1007/s12274-020-2852-3     

[2] Sangbong Lee et al 2020 Nanotechnology 31 095708, 10.1088/1361-6528/ab599c

[3] Lee et al., Nat Commun 3, 1024 (2012), 10.1038/ncomms2022

[4] Valeš et al. Sci Rep 10, 4516 (2020), https://doi.org/10.1038/s41598-020-60857-y

Reviewer 3 Report

In this work, the authors deposited a transfer support layer (TSL) on patterned sapphire substrate to facilitate the transfer of 3D graphene. Multilayer 3D Gr was grown on the TSL/PSS through Cu-vaporassisted CVD and successfully transferred onto a flexible substrate through wet etching without causing any structural damage. The authors fabricated flexible surface-enhanced Raman scattering (SERS) sensors using this flexible 3D Gr on PMMA, which could detect very low concentrations of R6G (down to 10-9 M). The present study is expected to provide a new strategy for the development of 3D Gr-based biosensors. I believe that publication of the manuscript may be considered only after the following issues have been resolved.

1.       In order to better highlight the advantages of this work, the author needs to provide a table to compare related work.

2.       Because SERS effect is a kind of interaction between light and matter. In order to better highlight its physical mechanism, it is suggested that the author supplement the UV-visible spectrum test of the sample.

3.       In Figure 4, the author needs to correct the peak position. Moreover, the enhancement factor needs to be calculated and analyzed quantitatively.

4.       The introduction can be improved. The articles related to some applications of graphene materials should be added such as Sensors 2022, 22, 6483; ACS Sustain. Chem. Eng. 2015, 3, 1677–1685; Diamond & Related Materials 128 (2022) 109273; Talanta 2015, 134, 435–442.

5.       Please check the grammar and spelling mistakes of the whole manuscript.

Reviewer 4 Report

In this article, Su Han Kim et al., developed a direct CVD method for the synthesis of transferable 3D graphene using an atomic-layer-deposited HfO2 layer as TSL layer. Few-layered 3D graphene can be conformally transferred onto flexible substrate through a conventional PMMA-assisted wet-etching transfer method. As a demonstration, the authors presented a sensitive and flexible surface-enhanced Raman scattering sensor based on the 3D graphene on PMMA with high detection performance for low concentration of R6G down to 10-9 M. This paper shall be publishable in Nanomaterials. Some concern or improvement shall be addressed for further improvement:

1.      A fundamental research goal in this work is to address the conformal transfer of 3D graphene onto other substrates. Thus several related key issues should be clarified or further interpreted.

(a)   The metal assisted growth can enable the growth of graphene on nearly arbitrarily substrate, if the key goal is compatible transfer, why the authors choose a chemically inert sapphire substrate, rather than a easily etchable substrate with no requirement of TSL?

(b)   I agree with the authors HfO2 is a good TSL layer, matching the graphene growth protocol in page 2 line 7-12, any specific requirement or supplementary date to elucidate the thickness influence on 3D graphene growth or transfer?

(c)   The Raman spectra in Figure 2e is a bit different from that on PSS substrate without HfO2 (https://doi.org/10.1016/j.carbon.2017.12.097) , HfO2 influences the growth of 3D graphene? Can the author explain this point?

2.      Compatible transfer is another key point in this work, however, there are at least two points contradict this clarification. First, the intensity of D peak, related with graphene defects state, increases after transfer (Figure 2e and Figure 3d); The AFM height profiles of 3D graphene are inconsistent (Figure 2d and Figure 2c). The author should interpret more.

3.      Figure 3b, the PMMA/Graphene can self-supported? This is a bit strange, I don’t quite sure whether it’s due to the 3D graphene support effect or PMMA support effect, the author can provide more details in method 2.2 section.

4.      Figure 4c and 4d, there are no graphene peaks in the sample? The author shall denote the main peak vibrations in Raman spectra.

5.      Silver nanoparticle shall be denoted in Figure 4a and 4b to avoid undesired misleading.

Round 2

Reviewer 1 Report

The raised questions and comments appear to be answered suitably in the rebuttal and reflected well in the revised manuscript. The reviewer recommends the publication of the manuscript.

Reviewer 3 Report

Accept in present form.

Reviewer 4 Report

The authors addressed all my concerns, I suggest the publication of this paper in this version.